# The microbiota protects from viral-induced neurologic damage through microglia-intrinsic TLR signaling

D Garrett Brown[1†], Raymond Soto[1†], Soumya Yandamuri[1], Colleen Stone[1], Laura Dickey[1], Joao Carlos Gomes-Neto[1], Elissa D Pastuzyn[2], Rickesha Bell[1], Charisse Petersen[1], Kaitlin Buhrke[1], Robert S Fujinami[1], Ryan M O'Connell[1], W Zac Stephens[1], Jason D Shepherd[2], Thomas E Lane[1]*, June L Round[1]*

[1]Department of Pathology, Division of Microbiology and Immunology, University of Utah School of Medicine, Salt Lake City, United States; [2]Department of Neurobiology, University of Utah School of Medicine, Salt Lake City, United States

**Abstract** Symbiotic microbes impact the function and development of the central nervous system (CNS); however, little is known about the contribution of the microbiota during viral-induced neurologic damage. We identify that commensals aid in host defense following infection with a neurotropic virus through enhancing microglia function. Germfree mice or animals that receive antibiotics are unable to control viral replication within the brain leading to increased paralysis. Microglia derived from germfree or antibiotic-treated animals cannot stimulate viral-specific immunity and microglia depletion leads to worsened demyelination. Oral administration of toll-like receptor (TLR) ligands to virally infected germfree mice limits neurologic damage. Homeostatic activation of microglia is dependent on intrinsic signaling through TLR4, as disruption of TLR4 within microglia, but not the entire CNS (excluding microglia), leads to increased viral-induced clinical disease. This work demonstrates that gut immune-stimulatory products can influence microglia function to prevent CNS damage following viral infection.
DOI: https://doi.org/10.7554/eLife.47117.001

*For correspondence:
tom.lane@path.utah.edu (TEL);
june.round@path.utah.edu (JLR)

†These authors contributed
equally to this work

Competing interests: The
authors declare that no
competing interests exist.

Reviewing editor: Peter
Turnbaugh, University of
California, San Francisco, United
States

## Introduction

Viral infection of the CNS can cause permanent neurologic damage and psychiatric disorders (*Bergmann et al., 2006*; *Koyuncu et al., 2013*; *van den Pol, 2009*). Indeed, multiple viruses including Zika and West Nile virus have been associated with Guillain-Barre syndrome and memory deficits, respectively (*Carson et al., 2006*; *Parra et al., 2016*; *Vasek et al., 2016*). Moreover, while Parkinson's disease, multiple sclerosis, Alzheimer's disease, narcolepsy and other chronic neurologic diseases have unknown etiologies, initiation by CNS viral infection has been implicated (*Itzhaki et al., 2016*; *Kakalacheva et al., 2011*; *Stoessl, 1999*; *Tesoriero et al., 2016*; *Virtanen and Jacobson, 2012*). Viral infection of the CNS presents unique challenges to the immune system with regard to controlling and eliminating the invading pathogen. Resident cells of the CNS are often targets for infection and are critical in induction of distinct pro-inflammatory signatures that function to attract virus-specific lymphocytes into the CNS. Consequently, a significant hurdle encountered by infiltrating antigen-specific lymphocytes is to eliminate virus from infected cells while limiting damage that may have long-term detrimental consequences to the host. Therefore, defining antiviral host defense mechanisms within the CNS might highlight novel therapeutic interventions to prevent or treat many of these viral-induced neurologic disorders.

Microglia are resident tissue phagocytes of the CNS that function in host defense and tissue homeostasis. Recent studies have shown that despite their location within the CNS, microglia are

**eLife digest** Trillions of bacteria, fungi and viruses live inside us, forming what is known as our microbiota. Far from causing problems, these microbes benefit our health in many ways. Most of our microbiota lives in our gut, yet there is increasing evidence that it can influence how our central nervous system works.

In particular, these communities of microbes could have a role in multiple sclerosis, a disease that emerges when the immune system attacks the insulating sheath which protects neurons, slowly leading to paralysis. What causes multiple sclerosis is still unknown, but scientists believe that a viral infection could trigger the condition. In the gut, the microbiota helps the immune system to fight off harmful microbes. It is still unclear whether it performs the same role in the central nervous system, and if it can participate in diseases where the immune system harms nerve tissues. Previous studies in mice have looked into how gut microbes influence the development of illnesses similar to multiple sclerosis, but they did not use the type of live viral infection that is thought to trigger the condition.

In rodents, a strain of mouse hepatitis virus (or MHV) causes symptoms similar to the ones observed in patients with multiple sclerosis: the animals become paralyzed and their neurons' protective sheaths get damaged. Brown, Soto et al. compared how mice that have their normal microbes, that were raised to be free of microbes, or that were given antibiotics responded to this virus. Animals that were germfree or had received antibiotics had weakened immune responses and failed to clear MHV. These mice also showed worse paralysis. Further experiments revealed that gut microbes protected against paralysis by switching on a cascade of molecular events in a specific type of immune cell in the nervous system. These findings suggest that in the central nervous system, the microbiota is critical to quickly clear viruses and to stop symptoms associated with multiple sclerosis from emerging.

Our own genetic background, but also lifestyle changes such as diets, antibiotics or sanitation can influence our microbiota. In parallel, in the past decades there has been an increase in the number of diseases, such as multiple sclerosis, in which the immune system turns against the body. The work by Brown, Soto et al. therefore emphasizes the need to maintain a diverse microbial community. For example, species of gut bacteria should be replaced or maintained after antibiotic treatments. However, future work is necessary to understand which of these microbes are protective, and whether they operate during specific timeframes.
DOI: https://doi.org/10.7554/eLife.47117.002

impacted by the gut microbiota (*Erny et al., 2015*; *Matcovitch-Natan et al., 2016*; *Sampson et al., 2016*). This is consistent with the known function of the microbiota on both immune cell composition and functionality (*Belkaid and Hand, 2014*; *Hooper et al., 2012*). Several studies have shown that resident commensal species can instruct the development of regulatory and inflammatory immune cell subsets that aid to limit viral replication at mucosal sites (*Abt et al., 2012*; *Robinson and Pfeiffer, 2014*). However, while the microbiota inhibits infection by some viruses, it can also enhance the infectivity of other viruses (*Robinson and Pfeiffer, 2014*). Thus, the contribution of the microbiota during viral infection may be distinct for each virus and might be determined by the location of infection. Despite advancements in this area, it remains unclear if microbial commensals impact either host defense and/or disease in response to CNS viral infection.

## Results

### The microbiota protects from neurologic damage associated with viral infection

To characterize the role of the microbiota in a model of viral-induced neurologic disease, 6- to 8-week-old C57BL/6 specific-pathogen-free (SPF) or germ-free (GF) mice were infected intracranially (i. c.) with the neurotropic JHM strain of mouse hepatitis virus (JHMV). JHMV infection results in an acute encephalomyelitis characterized by viral replication in astrocytes, microglia, and oligodendrocytes (*Bergmann et al., 2006*) (*Greenberg et al., 2014*) (*Blanc et al., 2014*; *Carbajal et al., 2010*; *Marro et al., 2016*; *Schrauf et al., 2008*; *Stiles et al., 2006*; *Chen et al., 2014*). Virus-specific T cells

control viral replication through secretion of anti-viral cytokines and cytolytic activity. Sterile immunity is not achieved and virus persists within white matter tracts in surviving mice that subsequently develop an immune-mediated demyelinating disease (*Bender and Weiss, 2010*; *Bergmann et al., 2006*; *Cheng et al., 2018*; *Lane and Hosking, 2010*; *Libbey et al., 2014*). As is typical in this model, infected SPF mice reach peak clinical disease by day 10–11 post-infection (p.i.) followed by a reduction of symptoms that plateaus around day 15–16 p.i. In contrast, GF mice developed worsened clinical disease that peaks at day 13 and is maintained through 21 days p.i. (*Figure 1A*). While GF mice had slightly lower titers of virus within the brain at 7 days p.i., control of viral replication was impaired at 14-days p.i. and virus was still present in GF mice at 21 days p.i., a time point where SPF animals have almost undetectable levels of infectious viral particles (*Figure 1B*). Importantly, GF mice displayed enhanced neurologic damage, as there was significantly more demyelination in the spinal cord when compared to SPF animals (*Figure 1C,D*). Immune cells traffic to the CNS in response to JHMV infection and viral clearance is dependent on both CD4 and CD8 T cells. Fewer total cells were recovered from the CNS of GF mice during peak viral infection (day 7) (*Figure 1E*). Moreover, there was a reduced frequency of total CNS-infiltrating CD4+ and CD8+ T cells (*Figure 1F,G*) and fewer viral-specific CD4+ and CD8+ T cells in GF mice at day 7 p.i. (*Figure 1H,I*). These differences are diminished at a chronic timepoint (day 21), as reductions in only total CD8 + and viral-specific CD8+ T cells are observed (*Figure 1G–I*). Inflammation is controlled by the presence of CD4+, FoxP3+ T regulatory (Tregs) cells and previous reports have shown a critical role for these cells in controlling neuroinflammation in response to JHMV infection (*Anghelina et al., 2009*). GF mice have increased CNS infiltrating Tregs during the acute phase of infection (7 p.i); however, there is a paucity of Tregs in infected GF mice during the chronic demyelinating phase of disease (*Figure 1J*).

The gut microbiota can be significantly reduced in the presence of antibiotics and is often used to validate findings in GF mice. To this end, progeny from breeder pairs of SPF mice that were orally treated with antibiotics, and remained on antibiotics after weaning, were subsequently infected i.c. with JHMV at 6–7 weeks of age. The same antibiotic regimen was performed previously, and qRT-PCR of bacterial 16S sequences demonstrated a reduction of fecal bacteria by approximately 100-fold (*Soto et al., 2017*). Similar to what was observed in GF mice, lifelong antibiotic-treated infected animals developed worsened chronic disease when compared to animals with an intact microbiota (*Figure 1—figure supplement 1A*) and were unable to control viral replication (*Figure 1—figure supplement 1B*). These data provide further support for the microbiota in functioning to coordinate disease following CNS viral infection.

## T cell-extrinsic defects from GF mice are responsible for worsened disease

Numerous reports have demonstrated a critical role for the microbiota in maturation of the immune response (*Belkaid and Hand, 2014*; *Hooper et al., 2012*). In particular, commensal microbes promote CD4+ T helper cell responses both within and outside of the intestine (*Atarashi et al., 2011*; *Ivanov et al., 2009*; *Lee et al., 2011*; *Mazmanian et al., 2005*). To determine whether increased disease in JHMV-infected GF mice is due to T cell-intrinsic or-extrinsic effects, either CD4+ or CD8 + T cells from immunized SPF or GF mice were adoptively transferred into JHMV infected Rag1$^{-/-}$ mice. SPF or GF mice were infected intraperitoneally (i.p.) with mouse hepatitis virus strain DM (MHV-DM) to generate virus-specific T cells (*Bergmann et al., 2004*; *Glass and Lane, 2003*). Splenic CD4+ or CD8+ T cells were purified and stained with tetramers to enumerate viral-specific T cells; subsequently, equal numbers of viral-specific T cells were separately transferred into JHMV infected Rag1$^{-/-}$ mice (*Figure 2A*). Consistent with published reports using other viruses, we observed slight systemic immune deficiencies in the response to i.p. infection with MHV-DM in GF mice in both total T cells and virus-specific T cells; only the number of viral-specific CD8+ T cells reached statistical significance (*Figure 2—figure supplement 1A–D*) (*Abt et al., 2012*; *Ichinohe et al., 2011*). Importantly, there were no differences observed in the recruitment of either CD4+ or CD8+ T cells derived from either GF or SPF mice to the CNS of JHMV-infected Rag1$^{-/-}$ mice, nor were there differences in CNS viral burden (*Figure 2B–G*). Collectively, these data show that neither CD4+ or CD8 + T cells from GF mice are intrinsically deficient in their ability to respond to JHMV or traffic to the CNS. Thus, these data suggest that the microbiota exerts effects on alternate cell types to protect from viral infection within the CNS.

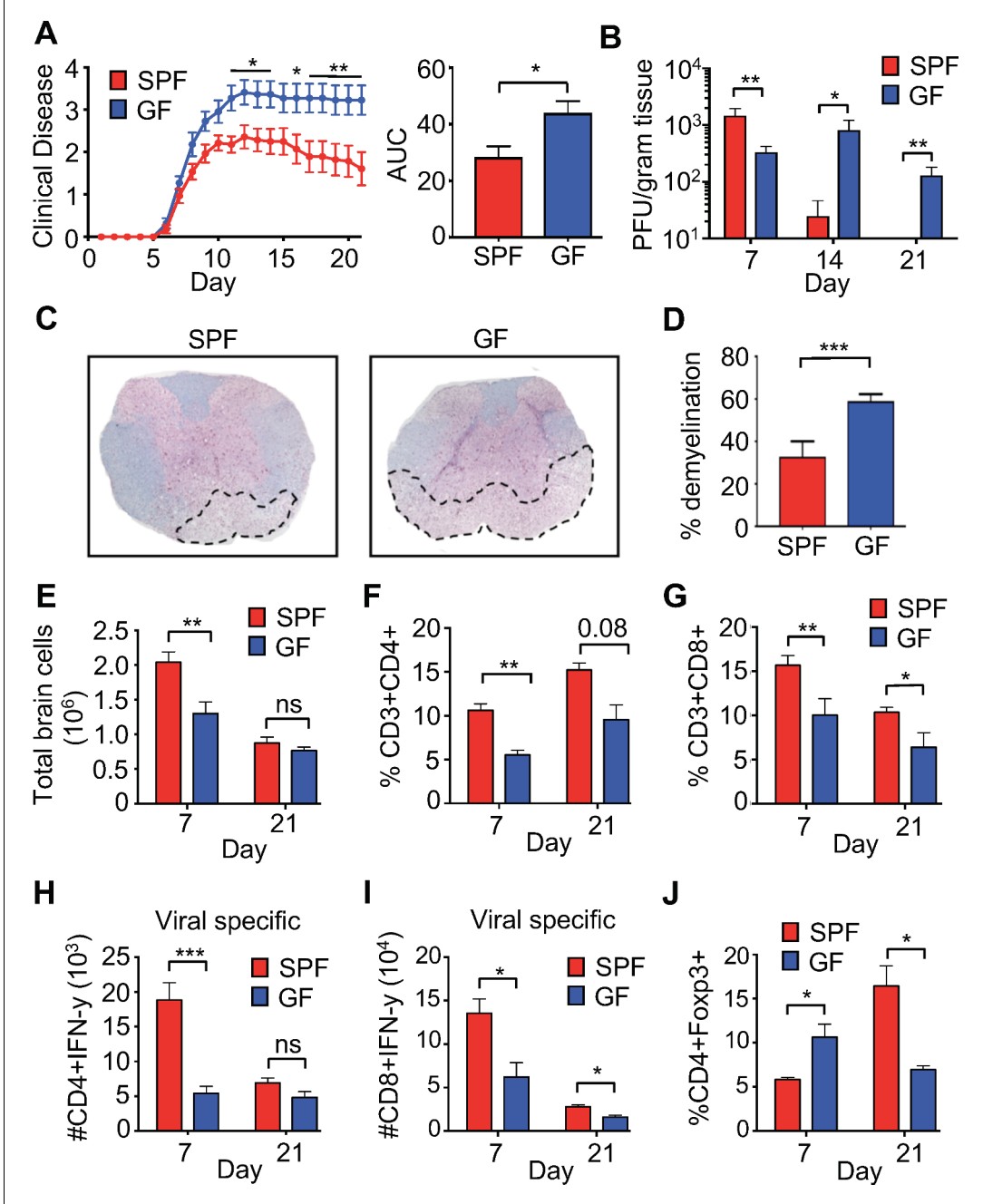

**Figure 1.** Perturbations to the microbiota leads to enhanced viral persistence and neuro-autoimmunity. (**A**) Clinical scoring (left) and the associated area under the curve (right) of SPF (n = 14) or GF C57BL/6 mice (n = 11) following intracranial (i.c.) infection with 150–300 PFU JHMV. Clinical disease was measured as described in the Materials and methods. Results shown from two independent experiments. (**B**) Supernatant from homogenized brains of JHMV infected SPF (n = 6–8) or GF mice (n = 6–9) at the indicated time points were used for determining viral titers. (**C**) Representative image of luxol fast blue/H&E stained spinal cords isolated from JHMV infected SPF or GF 21 days post-infection (p.i.), dashed lines highlight loss of myelin staining indicating demyelination. (**D**) Quantification of demyelination of SPF (n = 3) or GF (n = 3) mice at 21 days p.i. as represented in 1 c. (**E–J**) Brains from GF and SPF mice were analyzed via flow cytometry 7 and 21 days p.i. with 150 PFU of JHMV (n = 3–12). Results from one to two independent experiments. (**E**) Total number of brain cells isolated after homogenization of brain tissue and percoll gradient centrifugation. (**F**) CD3+, CD4 + frequency of recovered cells (**G**) CD3+, CD8+ frequency of recovered cells. Numbers of virus-specific CD4+ (**H**) and CD8+ (**I**) T cells as determined by intracellular IFN-γ staining in response to defined viral epitopes at indicated time points. (**J**) Foxp3+ frequency of CD4+ T cells recovered. All data displayed as mean with SEM. Means are of biological replicates (i.e. each data point is from a different mouse). Clinical score significance determined using two-way ANOVA statistical test with multiple comparisons. Viral titer significance determined by Mann-Whitney test. All other significance determined using Student's t-test. *p<0.05, **p<0.01, ***p<0.005.

*Figure 1 continued on next page*

*Figure 1 continued*

DOI: https://doi.org/10.7554/eLife.47117.003

The following figure supplement is available for figure 1:

**Figure supplement 1.** Reduced virus-specific immune responses within the CNS.

DOI: https://doi.org/10.7554/eLife.47117.004

## The microbiota upregulates antigen presentation in microglia

Microglia are the resident innate immune cells of the brain and are important for diverse processes including maintaining neuronal homeostasis, regulating brain development, and mounting immunity against pathogenic organisms. Commensal microbes are necessary for proper microglia maturation, with gene expression and morphological analysis suggesting that microglia isolated from GF mice have an immature phenotype (*Erny et al., 2015*; *Matcovitch-Natan et al., 2016*; *Sampson et al., 2016*). Yet, the effect of commensal microbes on microglial responses to CNS viral infection has yet to be well-tested. Flow cytometric analysis from homogenized brains followed by percoll gradient purification did not demonstrate significant differences in the frequencies of microglia from GF and SPF mice under homeostatic conditions (*Figure 3—figure supplement 1A–C*). One role for microglia is to activate antigen-specific T cell responses through presentation of antigens via MHC molecules (*Mack et al., 2003*). We identified a lower frequency of MHC class II + microglia in uninfected GF mice (*Figure 3A,B*). Disruption to the microbiota with antibiotics decreases both the percentage of total microglia and the number of MHC expressing microglia in uninfected animals (*Figure 3C–F*). Presentation of antigens requires co-stimulatory molecules such as CD86 and CD40 which are significantly reduced in expression on microglia from uninfected GF mice (*Figure 3G* and *Figure 3—figure supplement 1H*). CIITA is a transcription factor that is critical to the expression of MHCII expression (*Mach et al., 1996*). Expression of *Ciita* mRNA transcripts is significantly reduced in microglia isolated from GF or antibiotic-treated mice compared to SPF mice, further corroborating a reduction in the antigen presentation machinery in the absence of microbes (*Figure 3H,I*) (*Butovsky et al., 2014*). Three days post-infection, antigen presentation markers were also reduced in antibiotic-treated mice (*Figure 3—figure supplement 1D-G*), suggesting a weakened ability to activate T lymphocytes. To functionally validate these findings, we measured CD8+ T cell proliferation during coculture with microglia from either SPF or antibiotic-treated mice. Corroborating the reduced antigen presentation markers on microglia from antibiotic-treated mice, the ability of these microglia to induce T cell proliferation was reduced when presenting the immunodominant JHMV peptide S510–518 (*Figure 3J*).

## Microglia depletion worsens neurologic damage

Microglia are considered to be the first line of defense during a CNS infection. Indeed, recent studies have demonstrated that a reduction in microglia leads to a defect in control of JHMV replication within the CNS (*Wheeler et al., 2018*). However, the effect of microglia depletion on neurologic damage and demyelination remains unclear. To address this, we reduced the number of microglia using a colony-stimulating factor one receptor (CSFR1) inhibitor drug, PLX-3397 (*Elmore et al., 2014*). Administration of PLX-3397 1 week prior to JHMV infection of SPF mice resulted in a marked reduction in microglia (CD45lo, CD11b+, F4/80+ cells as determined by flow cytometric staining) in both number and percentage at days 7 and 14 p.i. (*Figure 4A,B*). Recent studies have demonstrated that treatment of West Nile Virus-infected mice with a CSF1R inhibitor results in limited viral control associated with impaired activation of antigen presenting cells (APCs) in both blood and draining lymph nodes arguing that in addition to targeting microglia other immune cells are also depleted leading to a potentially immunosuppressed state (*Funk and Klein, 2019*). Similar to this study, we did not observe a reduction in either splenic (CD45+, CD11b+) or CNS-infiltrating macrophage (CD45hi, CD11b+) populations (*Figure 4—figure supplement 1A*; *Figure 4—figure supplement 1B*; *Figure 4—figure supplement 1C*). We observed enhanced mortality in JHMV-infected mice following microglia reduction that correlated with an impaired ability to control viral replication within the CNS (*Figure 4C,D*). Most importantly, microglia depletion leads to increased demyelination compared to control mice (*Figure 4E,F*). Similar to what we observe in GF mice, both CD4+ and CD8+ T cell responses are deficient in the absence of microglia and IFN-γ production from these

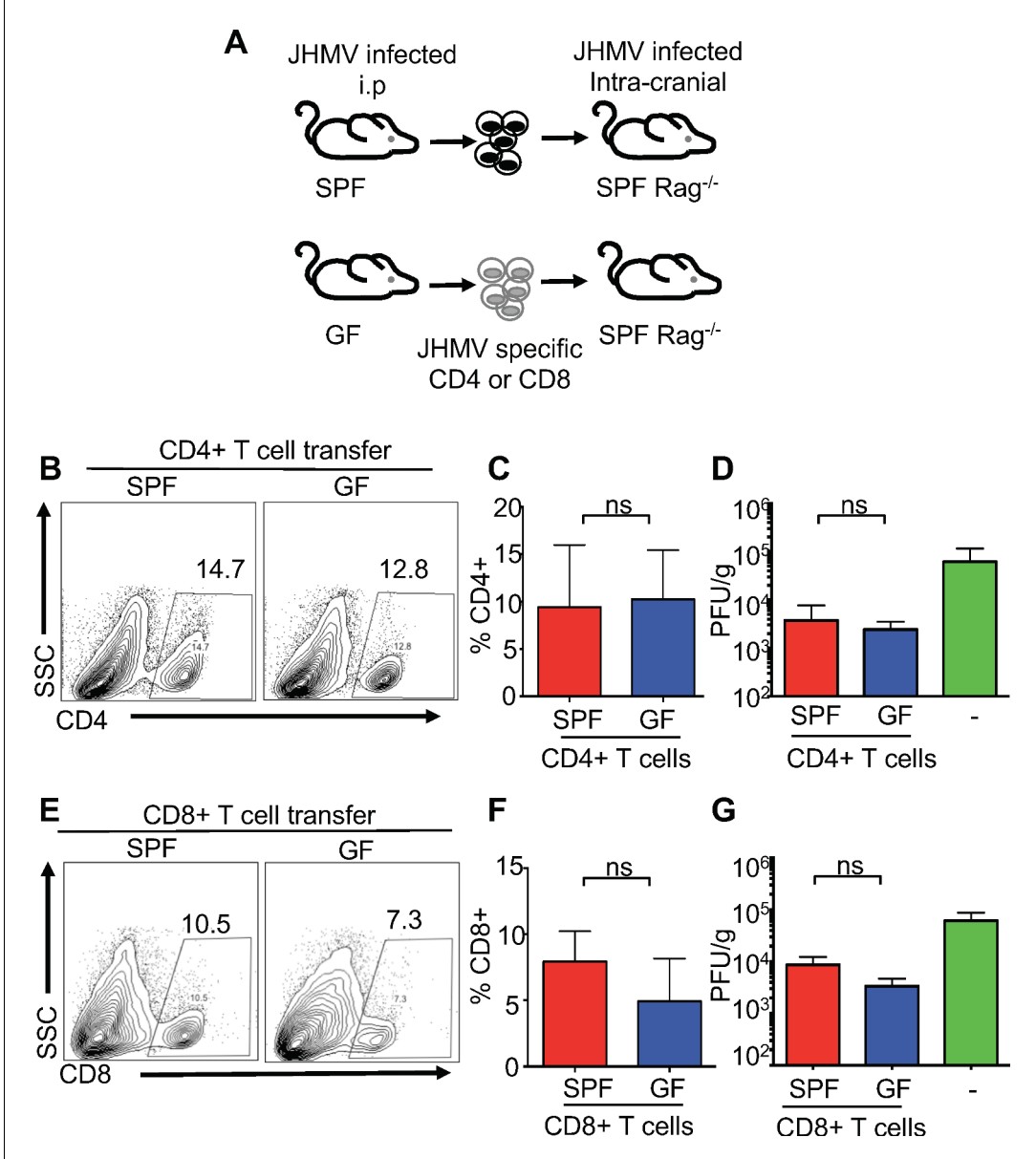

**Figure 2.** T cell-intrinsic defects are not responsible for reduced viral clearance during CNS infection in GF mice. (**A**) Schematic of experimental design; SPF or GF mice were intraperitoneally (i.p.) infected with the hepatotropic MHV-DM and CD4+ or CD8+ T cells were isolated from donor spleens at day 7 p.i. Subsequently, $2.5 \times 10^5$ virus-specific T cells (determined via tetramer staining) were retro-orbitally injected into SPF Rag1$^{-/-}$ and recipient mice that were i.c. infected with JHMV (150 PFU) three days prior to cell transfer; recipient mice were subsequently sacrificed 7 days post-transfer to determine viral titers within the brains. (**B–C**) Flow cytometry analysis and percentages of CD4+ T cells from CNS of JHMV-infected Rag$^{-/-}$ mice (n = 7). (**D**) Viral titers in brains of JHMV-infected Rag1$^{-/-}$ mice receiving virus-specific CD4+ T cells from either SPF or GF mice or vehicle alone (n = 4–7). (**E and F**) Flow cytometry analysis and percentages of CD8+ T cells from CNS of Rag$^{-/-}$ mice given viral-specific CD8+ T cells and infected with 150 PFU of JHMV. n = 5–6. (**G**) Viral titers in brains of JHMV-infected Rag1$^{-/-}$ mice receiving virus-specific CD4+ T cells from either SPF or GF mice or vehicle alone. Significance determined by Student's t-test. All data displayed as mean with SEM. Means are of biological replicates.

DOI: https://doi.org/10.7554/eLife.47117.005

The following figure supplement is available for figure 2:

**Figure supplement 1.** Reduced immune responses in GF mice are not a result of T cell-intrinsic defects.

DOI: https://doi.org/10.7554/eLife.47117.006

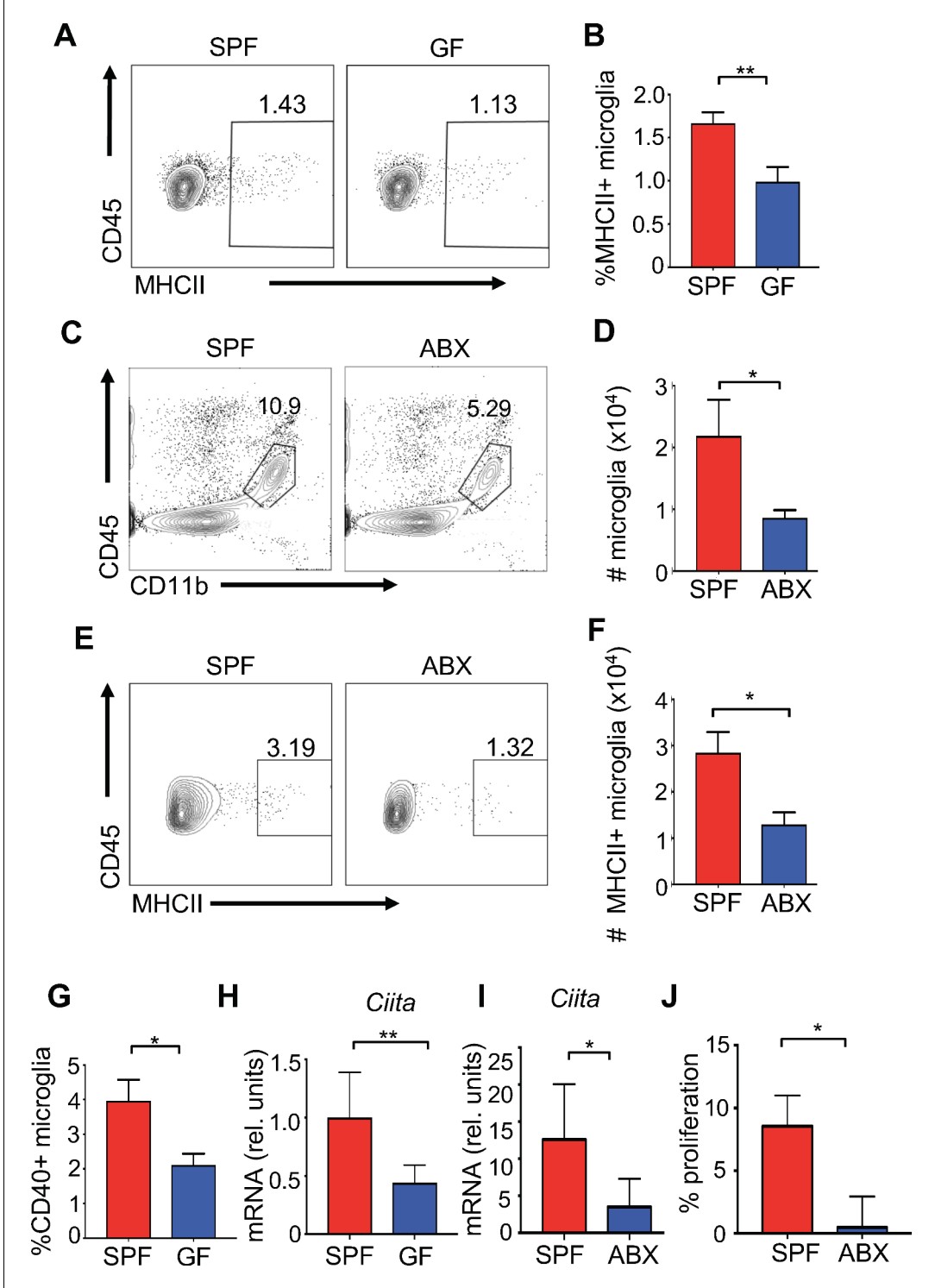

**Figure 3.** The microbiota promotes antigen presentation by microglia. (**A**) Representative flow plots showing microglia isolated from brains of uninfected SPF and GF mice; (**B**) Quantitation from multiple mice from (**A**). (**C**) Representative flow plots of CD45lo, CD11b+, F480+ cells isolated from the brain of antibiotic treated (ABX) and SPF mice. (**D**) Quantitation from multiple mice from (**C**). (**E**) Representative flow cytometric analysis of MHC class II expression on microglia from uninfected SPF and ABX mice; (**F**) quantification of frequency of MHC class II positive microglia from (**E**). (**G**) Microglia from uninfected SPF and GF mice were evaluated for CD40 by flow cytometry. (**A–G**) Results are combined from two to three independent experiments (n = 3–9/experiment). (**H**) qPCR of *Ciita* from purified microglia from SPF or GF mice at day 3 p.i. with JHMV (n = 5–6). (**I**) qPCR of *Ciita* from FACS purified microglia from ABX or SPF mice at day 3 p.i. with JHMV (n = 3–4). (**J**) Quantification of proliferation of CFSE labeled JHMV specific
*Figure 3 continued on next page*

*Figure 3 continued*

CD8+ T cells incubated with microglia from antibiotic treated mice (n = 5), pulsed with viral peptide. Data from two independent experiments and represented as mean. All data displayed as mean with SEM. All means are from biological replicates. All significance determined by Student's t test. *p<0.05, **p<0.01.

DOI: https://doi.org/10.7554/eLife.47117.007

The following figure supplement is available for figure 3:

**Figure supplement 1.** Perturbations to the microbiota lead to reduced activation of microglia.

DOI: https://doi.org/10.7554/eLife.47117.008

cells is impaired (*Figure 4G–I* and *Figure 4—figure supplement 1D*; *Figure 4—figure supplement 1E*; *Figure 4—figure supplement 1F*; *Figure 4—figure supplement 1G*; *Figure 4—figure supplement 1H*). Collectively, these data support a role for microglia for limiting neurologic damage.

## Oral administration of TLR ligands is sufficient to enhance microglia activation and protect from neurologic disease

Previous reports have identified that short chain fatty acids (SCFAs) are able to influence the morphology and transcriptome of microglia (*Erny et al., 2015*); however, the microbiota is also a rich source of microbial ligands that can bind to Toll-like receptors (TLRs) that are known to influence systemic immune responses (*Kubinak and Round, 2012*). Furthermore, the microbiota is also known to induce several cytokines that have the potential to influence microglial function (*Pinteaux et al., 2002*). To identify relevant microbial products that might directly activate microglia, we screened various SCFAs, TLR ligands and innate cytokines on BV2 cells to evaluate induction of MHC expression. The BV2 cell line was originally developed by infecting primary microglial cell cultures with a -raf/v-myc oncogene carrying retrovirus (J2) (*Blasi et al., 1990*) and has been used by numerous investigators as an effective alternative model system for primary microglia cultures to study various cell biology aspects (*Henn et al., 2009*). While SCFAs were able to induce MHCII expression at very high concentrations, lower levels of most SCFAs and cytokines associated with the innate immune response, for example IL-1, IL-18 and IL-33, did not stimulate MHCI and II expression in BV2 cells (*Figure 5A,B* and *Figure 5—figure supplement 1A–C*). Butyrate was able to stimulate a small, but significant, increase in MHC expression; however, the highest levels of MHCI and II upregulation were achieved by bacterial-associated TLR ligands (*Figure 5A,B* and *Figure 5—figure supplement 1C*). LPS (and to a lesser extent, the TLR1/2 heterodimer ligand Pam3CysK4) significantly upregulated both the frequency of MHCII expressing cells and the total amount of MHCI per cell (*Figure 5A,B* and *Figure 5—figure supplement 1C*). These data suggest that TLR ligands from the microbiota are able to prime microglia for antigen presentation.

To test if microbial ligands could increase antigen presentation machinery on microglia in vivo, GF mice were orally fed either the TLR4 ligand LPS or LPS in combination with the TLR1/2 ligand Pam3CysK4 to replicate the natural route of exposure to microbiota-produced ligands. Animals were fed TLR ligands for 2 weeks, and subsequently infected with JHMV. The level of TLR ligands fed in these experiments does not lead to excessive levels of TLRs in the serum and does not exceed those found in SPF animals (*Figure 5—figure supplement 2A,B*). Feeding GF mice bacterial cell wall components resulted in reduced paralysis and lower viral titers at day 9 p.i. (*Figure 5C,D*). Consistent with this, increases in cell number, CD4 and CD8 T cells in the CNS of TLR ligand fed animals was also observed (*Figure 5—figure supplement 2C–H*). Feeding GF animals LPS alone, led to similar increases in cell number in the CNS of infected mice (*Figure 5E*), more MHCII expressing microglia and increased CD4 T cell trafficking to the CNS during infection (*Figure 5F,G*). Interestingly, feeding mice both ligands increased numbers and frequency of CD4 and number of CD8 T cells in the CNS (*Figure 5—figure supplement 2D–H*), while LPS only increased total numbers of CD4 T cells (*Figure 5G* and data not shown). Feeding mice LPS alone, however, increased MHCII expression on microglia during disease (*Figure 5F*), but this was not observed when feeding both LPS and Pam3Cysk4 (data not shown). These results demonstrate that TLR ligands derived from the gut microbiota are sufficient to influence microglial activation and protection in response to viral-induced neurologic disease.

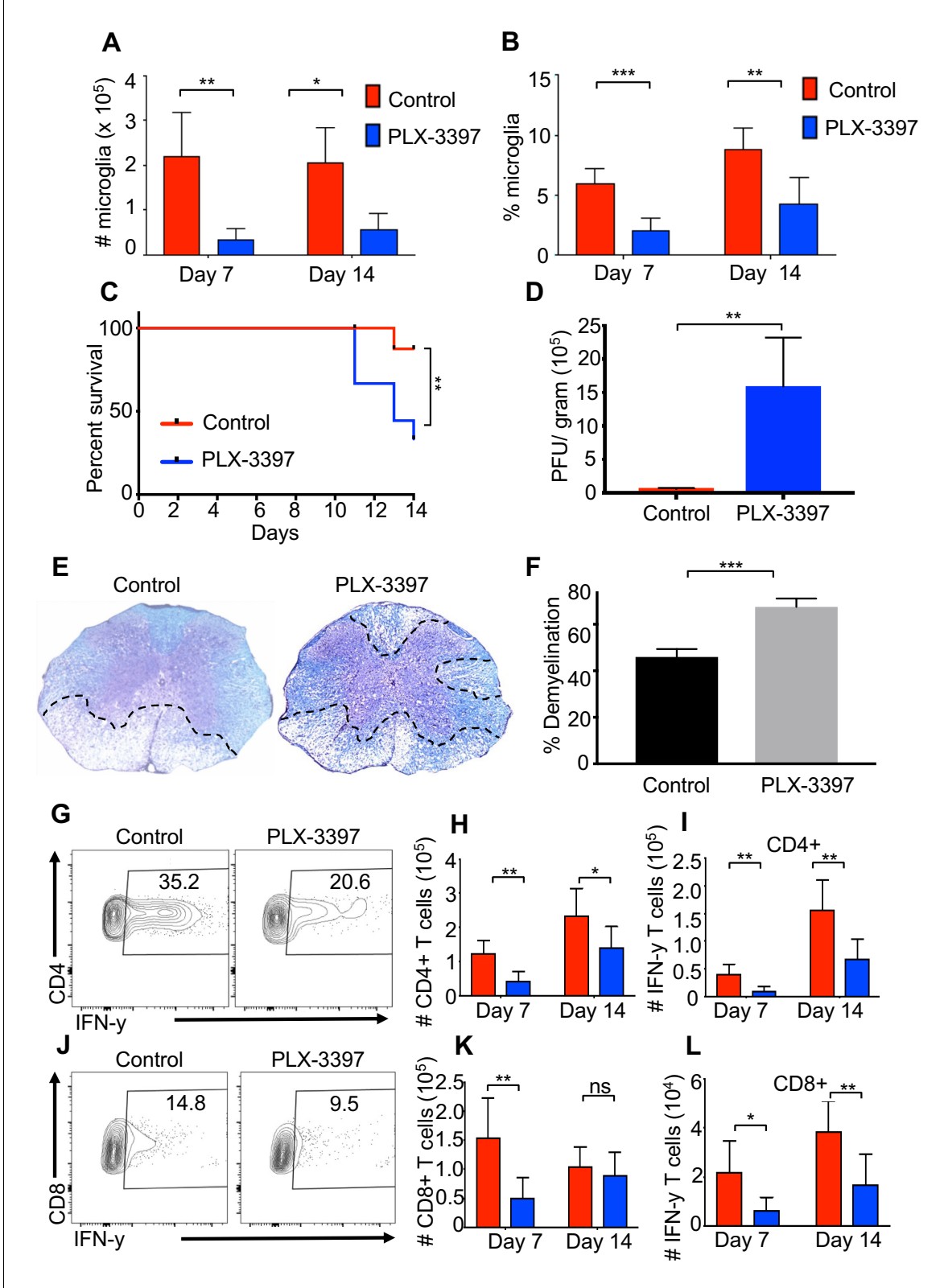

**Figure 4.** Microglia are beneficial in limiting JHMV-induced demyelination. SPF mice were fed chow containing PLX-3397 7 days prior to i.c. infection with 150 PFU JHMV. (A) Numbers and (B) frequency of microglia as determined by flow cytometric analysis at days 7 and 14 p.i. (n = 5–9). (C) Survival of microglia depleted and control mice after infection. JHMV-infected mice surviving out to day 14 p.i. were sacrificed and viral titers (D) and the severity of spinal cord demyelination determined (D–F). (E) Representative LFB-stained spinal cords illustrate demyelination (outlined with black hashed lines) in

*Figure 4 continued on next page*

*Figure 4 continued*

PLX-3397-treated mice compared to control mice. (**F**) Quantification of the severity of spinal cord demyelination a represented in 4e. (**G**) Representative flow plots of IFN-γ+, CD4 T cells in the brains of control or PLX-3397 fed mice at day 7 p.i. (**H–I**) Quantification of CD4 T cell number and IFN-γ+ CD4 T cell number at days 7 and 14 p.i. in control and PLX-3397 fed mice. (**J**) Representative flow plots of IFN-γ+, CD8 T cells in the brains of control or PLX-3397 fed mice at day 7 p.i. (**K–L**) Quantification of CD8+ T cell number and IFN-γ+, CD8 T cell number at days 7 and 14 p.i. in control and PLX-3397 fed mice. IFN-γ was induced by restimulation with ionomycin and PMA. Survival curve significance determined by using the logrank test. All other significance determined by Student's t test. Survival curve is displayed as percent survival within each group. All other data displayed as mean with SEM. Means are of biological replicates. *p<0.05, **p<0.01.

DOI: https://doi.org/10.7554/eLife.47117.009

The following figure supplement is available for figure 4:

**Figure supplement 1.** Microglia depletion leads to reduced inflammatory T cell infiltration.

DOI: https://doi.org/10.7554/eLife.47117.010

## *Tlr4*, but not *Tlr2*, is required for protection from viral-induced neurologic damage

Our results demonstrate that gut microbiota products influence priming of resident cells of the CNS through TLR-dependent signals. Feeding TLR ligands to GF mice suggests that these microbial products are sufficient to decrease neurologic disease, and our results show that TLR4 exposure may play a more prominent role than TLR2 signaling in this process. A previous report demonstrated that there is no difference in the morphology of microglia from TLR deficient mice (*Erny et al., 2015*), but antigen presentation in these mice was not evaluated in these studies. Therefore, we infected TLR2$^{-/-}$ and TLR4$^{-/-}$ animals with JHMV and assayed for neurologic damage. Compared to WT mice, TLR2$^{-/-}$ animals had no difference in mortality (**not shown**) or clinical presentation of disease (*Figure 6—figure supplement 1A*). TLR4$^{-/-}$ mice had significantly worsened disease symptoms that were similar to GF mice (*Figure 6A*). Importantly, TLR4$^{-/-}$ animals had significantly increased demyelination, indicating that TLR4 signals are important for protection from neurologic damage (*Figure 6B, C*). Consistent with this, microglia, but not macrophages, from TLR4$^{-/-}$ animals express lower levels of MHCII under homeostatic conditions, prior to infection (*Figure 6D,E* and *Figure 6—figure supplement 1B*) and lower total numbers of MHCII expressing microglia (and total microglia) during JHMV infection when compared to WT mice (*Figure 6F and G*). JHMV is unable to directly activate TLR4 (*Figure 6—figure supplement 1C*), further arguing that microbiota-based signals function to prime microglia against CNS infection. Collectively, these results argue that brain resident immune cells are specifically influenced by TLR4 signals to protect from viral-induced CNS damage.

## Microglia-intrinsic TLR4 signaling is required for protection from viral-induced paralysis

Collectively, these data argue that gut microbial products are important for priming microglia within the CNS. However, these experiments cannot differentiate between TLR4 signals exerting activity within the gut or by directly priming microglia themselves. Indeed, we and others have detected TLR4 agonists within the blood of mice and humans, suggesting that microbial products could be circulating within the blood and thus be available to prime cells at sites distant from mucosal tissue (*Clarke et al., 2010*; *Soto et al., 2017*). One study, however, has suggested that LPS is unable to cross the blood-brain barrier; therefore, it is unclear how TLR4 signals derived from the gut microbiota could influence microglia within the CNS (*Banks and Robinson, 2010*). To initially address this, we performed a bone marrow chimera experiment. Radiation of animals followed by a bone marrow transplant readily replaces cells of the hematopoietic system in most compartments; however, microglia are known to be radio-resistant and therefore, during these experiments the majority of microglia are still from the recipient animal. Based on this, we performed bone marrow chimeras where we transplanted either WT or TLR4$^{-/-}$ bone marrow into WT recipients. Similarly to our previous results, we find that TLR4$^{-/-}$ animals develop significantly worsened clinical disease compared to WT animals (*Figure 7A*). Moreover, animals receiving either WT or TLR4$^{-/-}$ bone marrow, where most of the systemic hematopoietic system is replaced by TLR4 deficient cells, still develop disease similar to WT animals (*Figure 7A*). To address this using a genetic approach, we crossed TLR4-floxed animals to either a nestin-Cre (CNS$^{\Delta TLR4}$) driver or an inducible CX3CR1$^{CreER}$ driver (CX3CR1$^{\Delta TLR4}$). Nestin is expressed in neural progenitor cells that give rise to neurons, oligodendrocytes and astrocytes of

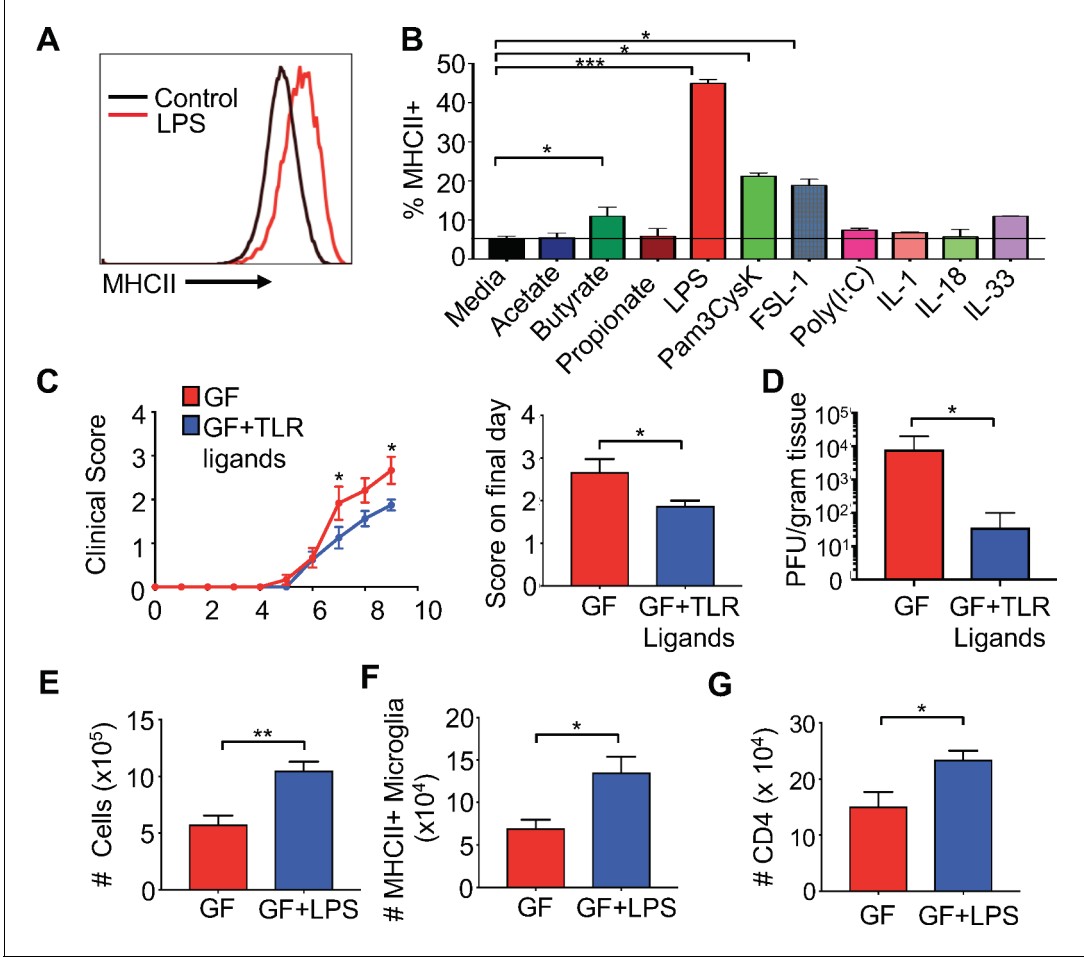

**Figure 5.** Gut administration of TLR ligands is sufficient to promote microglia antigen presentation and protection from neuro-autoimmune disease. (**A**) Representative flow cytometric analysis of MHC class II staining on BV2 microglial cell lines treated with LPS. (**B**) Quantification of flow cytometric analysis of MHC class II + BV-2 cells treated with SCFAs, TLR ligands (at 0.1 µg/mL), or indicated cytokines (0.25 µg/mL); data presented as mean with SEM and represent four independent experiments. Means are of technical replicates (i.e. multiple aliquots from the same cell line were tested for response to each ligand). (C to K) GF (n = 12) and GF mice with *ad libitum* access to drinking water containing Pam3CysK4 and LPS (n = 5) or LPS alone (n = 3) were i.c. infected with 150 PFU JHMV. (**C**) Disease scores and (left) and score on final day (right) and final day brain viral titer (**D**) of JHMV-infected mice. Number of cells (**E**) and MHCII+ microglia (**F**) on day 9 of infection. (**G**) Number of CD4+ T cells from brains of GF mice or GF mice fed LPS. Error bars represent SEM. Means are calculated from biological replicates. Clinical score significance determined using two-way ANOVA statistical test. All other significance determined by Student's t test. *p<0.05, **p<0.01, ***p<0.005.

DOI: https://doi.org/10.7554/eLife.47117.011

The following figure supplements are available for figure 5:

**Figure supplement 1.** TLR ligands upregulate antigen presentation machinery with a microglia cell line.
DOI: https://doi.org/10.7554/eLife.47117.012

**Figure supplement 2.** Oral treatment with TLR ligands is sufficient to increase the immune response during CNS viral infection.
DOI: https://doi.org/10.7554/eLife.47117.013

the CNS; however, its expression in microglia has been contentious. To verify that Nestin is not expressed in microglia, we crossed the nestin-Cre driver to a Rosa-lox-stop-lox-eYFP animal, allowing us to fate map nestin (and thus YFP) expressing cells. While we could readily detect YFP expression within the CNS, we were unable to detect YFP+ cells in the microglia fraction (*Figure 7—figure supplement 1A*), further corroborating that nestin-Cre will not drive recombination in microglia. CX3CR1 is expressed in microglia, but also in other immune cells such as dendritic cells in the gut; however, since microglia are long-lived, the use of an inducible system allows for depletion of a gene within microglia, while other immune cells that might be affected by the tamoxifen treatment

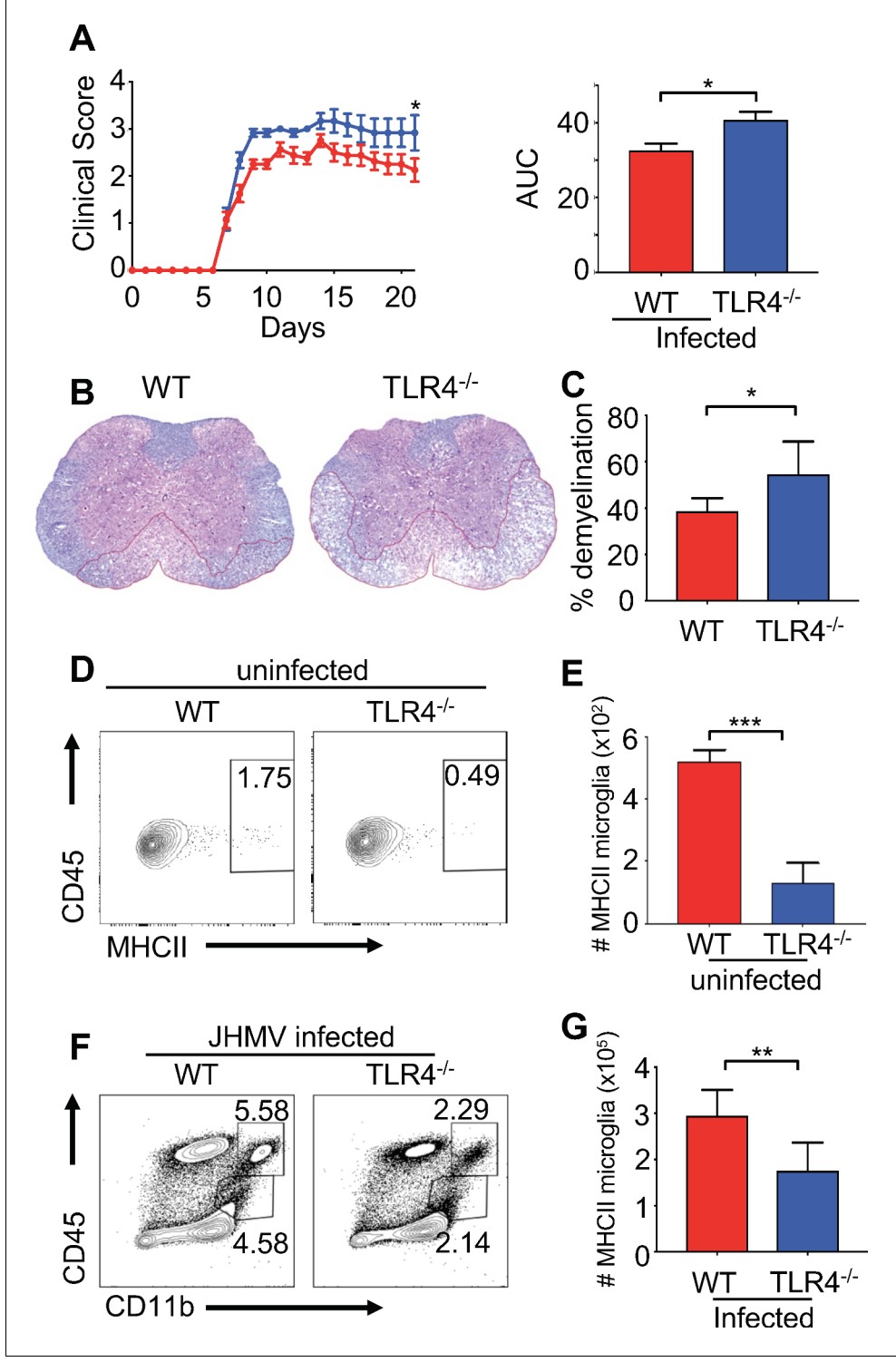

**Figure 6.** TLR4 signaling protects from JHMV-induced neurologic disease. WT C57BL/6 (n = 8) and TLR4⁻/⁻ (n = 6) were infected i.c. with 150 PFU of JHMV and followed for 21 days. (**A**) Clinical scores (left) and associated area under curves (right) from infected mice. (**B**) Representative luxol fast blue/H&E staining of spinal cord sections, with lines illustrating demyelination of WT and TLR4⁻/⁻ mice at 21 days p.i. (**C**) Quantification of spinal cord demyelination of WT (n = 7) and TLR4⁻/⁻ (n = 4) mice at 21 days p.i. Representative flow plot (**D**) and total number of MHC class II + microglia (**E**) from WT and TLR4⁻/⁻ mice under steady state conditions (n = 5). (**F–G**) WT and TLR4-/- (n = 8) mice were infected with JHMV and analyzed at day 9 p.i. Representative flow plots of microglia (**F**),
*Figure 6 continued on next page*

*Figure 6 continued*

total number of MHCII+ microglia (**G**). All data displayed as mean with SEM. Means are of biological replicates. All significance determined by Student's t test. *p<0.05, **p<0.01, ***p<0.001.

DOI: https://doi.org/10.7554/eLife.47117.014

The following figure supplement is available for figure 6:

**Figure supplement 1.** TLR4 signals within microglial cells to limit morbidity in response to JHMV infection of the CNS.

DOI: https://doi.org/10.7554/eLife.47117.015

will rapidly turn over and revert back to a WT status (*Goldmann et al., 2013*). This is currently the most widely used method to knock genes out within microglia. We verified that this treatment lead to disruption of TLR4 within microglia (*Figure 7—figure supplement 1B*). Based on this, CX3CR1$^{\Delta TLR4}$ animals were tamoxifen treated and subsequently infected 1 month post-tamoxifen treatment. While infection of CNS$^{\Delta TLR4}$ animals did not demonstrate a difference in clinical disease compared to WT animals (*Figure 7B*), disruption of TLR4 specifically within microglia, lead to worsened paralysis (*Figure 7C*). Collectively, these data demonstrate that gut microbial signals can directly influence microglia function to prevent viral-induced neurologic diseases.

## Discussion

Consistent with the known contribution of the microbiota to immune system development, studies performed almost 50 years ago suggested that resident commensals aid in protection from influenza virus (*Dolowy and Muldoon, 1964*). However, more recent studies have uncovered that other viruses take advantage of the microbiota to enhance infectivity (*Robinson and Pfeiffer, 2014*). Many human neurotropic viral pathogens are associated with the development of neurologic dysfunction (*Koyuncu et al., 2013*), yet little is known about the functional contribution of bacterial symbionts in host defense and disease following infection of the CNS. We demonstrate that products from the microbiota are sufficient to prime microglia within the CNS that aid in control of viral replication through microglia-intrinsic TLR4 signaling. Reductions in microbial stimulation through increased antibiotic use and sanitation, has been suggested to contribute to the significant increase in obesity, autoimmunity and even some neurologic disorders in the western world (*Blaser, 2016*; *Strachan, 2000*). Supporting this hypothesis, our findings demonstrate that loss of immune stimulation by the gut microbiota leads to failure to control viral replication within the CNS leading to enhanced neuropathy. Thus, loss of protective resident microbes can lead to CNS dysfunction.

There is an emerging interest in how gut commensals can influence diverse processes within the nervous system (*Fung et al., 2017*). The microbiota influences serotonin production, microglia gene expression patterns, and disease pathology in a mouse model of Parkinson's disease (*Erny et al., 2015*; *Reigstad et al., 2015*; *Sampson et al., 2016*; *Yano et al., 2015*), yet little is known about the mechanisms by which the microbiota govern these processes. Recent studies have demonstrated that SCFAs are able to influence the gene expression pattern and activation state of microglia (*Erny et al., 2015*); however, it is likely that other molecules exist that could influence the maturation of the CNS. We identify that bacterial cell wall products, such as LPS, are sufficient to prime microglia for antigen presentation to effectively clear virus. While microglia have long been known to express TLRs, this family of receptors has been primarily thought to function only during infection. Here, we show that the microbiota regulates microglia function through TLR4, priming these cells to respond to infection. Microglia develop early in embryogenesis from yolk sac progenitors; however, in contrast to macrophages, microglia are long-lived without any significant input from circulating blood cells (*Prinz et al., 2014*). How TLR ligands signal to cells within the CNS is unclear, however, there is evidence that gut microbial products are found circulating within the blood and could reach the CNS through this route (*Clarke et al., 2010*). This requires further investigation as some reports have shown that LPS is not able to penetrate the blood–brain barrier. While our data demonstrate that TLR4 signaling by microglia is, in part, responsible for orchestrating microglia activation, the possibility exists that gut microbiota signals can be transmitted to the CNS from the enteric nervous system. Indeed, enteroendocrine cells have been reported to contain neuropods that are directly linked to neuronal cells and are able to transmit signals to the CNS (*Bohórquez et al., 2015*). It

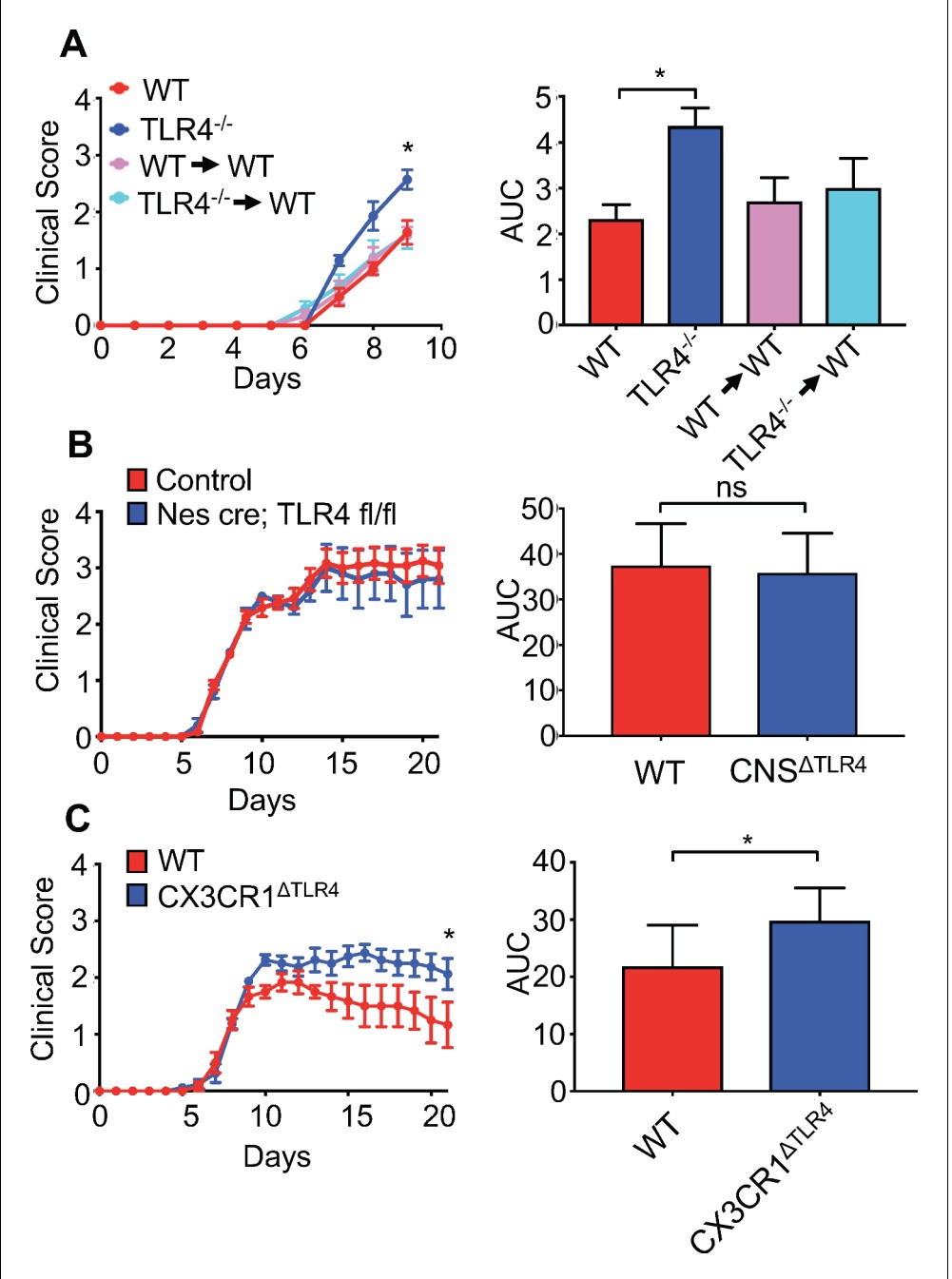

**Figure 7.** Microglial-specific TLR4 signaling protects from neurologic disease. (**A**) WT, TLR4$^{-/-}$, WT mice receiving WT bone marrow and WT mice receiving TLR4$^{-/-}$ bone marrow were infected with JHMV and clinical scores (left) and area under curve (right) calculated. (**B**) Nestin-Cre or TLR4-fl/fl (as WT controls) (n = 12) and nestin-Cre TLR4-fl/fl (n = 5) were infected with JHMV and clinical scores (left) and area under curve (right) determined. Results representative of two independent experiments (**C**) TLR4 fl/fl (n = 6) (as WT controls) and CX3CR1$^{CreER}$ TLR4 fl/fl (n = 8) mice were treated with tamoxifen, and 1 month later infected with 150 PFU of JHMV. Clinical scores (left) and area under curve were calculated. All data displayed as mean with SEM. Means are of biological replicates. Clinical score significance determined using two-way ANOVA statistical test with multiple comparisons. Area under the curve significance calculated by one-way ANOVA, comparing each group to WT (**B**) or by Student's t test (**C**). *$p<0.05$, ***$p<0.001$.

DOI: https://doi.org/10.7554/eLife.47117.016

The following figure supplement is available for figure 7:

*Figure 7 continued on next page*

*Figure 7 continued*

**Figure supplement 1.** TLR4 signals within microglial cells to limit morbidity in response to JHMV infection of the CNS.

DOI: https://doi.org/10.7554/eLife.47117.017

should also be noted that many of our experiments do not differentiate effects of the gastrointestinal microbiota from commensals occupying other niches. Our data from orally administered LPS likely limits effects to gastrointestinal exposure, but oronasalpharyngeal and pneumonic exposure may be occurring as well. Relatedly, differences observed between feeding mice LPS alone, and in combination with Pam3CysK4 suggest that more research could be performed investigating the interaction between multiple TLR ligands on microglia function and response to infection. Moreover, we cannot completely rule out a contribution from gut-resident CX3CR1 cells to this phenotype or other migrating DC populations. Recent studies have demonstrated that cells from the gut can migrate to the brain and there exist a population of long-lived CX3CR1 cells within the gut (*Shaw et al., 2018*). As these gut cells are still radio-sensitive (microglia are not), our bone marrow chimera studies argue, that these cells likely only play a minor role, if any. Future investigations using animal models that allow tracking of specific populations from the gut will aid in parsing out these details. Therefore, additional work remains to be performed to identify how gut microbes influence distal CNS processes.

The clinical implications of this work further support that maintenance of a complex microbial community is important to protect from diseases that plague the western world. Indeed, there are several examples of cell wall molecules from the microbiota that are sufficient to treat disease in various animal models (*An et al., 2014*; *Mazmanian et al., 2008*; *Vatanen et al., 2016*; *Wang et al., 2014*). Our findings support that immunostimulatory LPS is one of these microbial cell wall products. There are several different types of LPS that are derived from the microbiota, indeed LPS from *Bacteriodes* is less stimulatory than LPS derived from *E. coli* (*Vatanen et al., 2016*); therefore, the composition of the microbiota and the type of immune stimulatory products being delivered by the microbiota likely dictate maturation of microglia within the CNS. Additionally, fatty acids that can be produced by the microbiota are also capable of binding TLR4, and might constitute other relevant agonists (*Wong et al., 2009*). As microglia are also known to influence the function of neurons, these results suggest that microbial control of microglia could influence neuronal activity and function (*Parkhurst et al., 2013*). Much remains to be explored regarding the mechanisms by which gut microbes influence the brain; however, our findings highlight that appropriate microbial stimulation is critical to steady state development of the CNS and optimum neurologic health.

## Materials and methods

### Mice

SPF C57BL/6J, birth antibiotic C57BL/6J, *Rag*1$^{-/-}$, nestin-Cre, *Il1r*$^{-/-}$, *Il18*$^{-/-}$, *Tlr4*$^{-/-}$, TLR4-floxed, *Cx3xr*$^{CreER}$, and *Tlr2*$^{-/-}$ mice were originally purchased from Jackson Labs and reared in the SPF facility at the University of Utah. Germfree C57BL/6J mice were reared in gnotobiotic chambers in the germfree facility at the University of Utah. Microbial sterility is determined every 3 weeks by plating and PCR. Mice used in this study were 6–10 weeks old at time of infection. For microglia depletion, PLX-3397 (formulated in an AIN-76A) was prepared by Research Diets and provided by Plexxikon Inc SPF mice were provided PLX-3397 in their chow beginning 7 days prior to viral infection. Mice were chosen at random to receive either PLX-3397 or control chow. All experiments were performed in accordance to federal regulations as well as the guidelines for animal use set forth by the University of Utah Institutional Animal Care and Use Committee.

### Viral infection

Mice were intra-cranially infected with 150–600 plaque forming units (PFU) of the neurotropic JHMV strain J2.2v-1 suspended in 30 μL of PBS and clinical disease monitored using an established scoring sysem previously described (*Blanc et al., 2014*; *Carbajal et al., 2010*; *Chen et al., 2014*; *Greenberg et al., 2014*; *Marro et al., 2016*; *Schrauf et al., 2008*; *Stiles et al., 2006*). Germ-free

mice were maintained on antibiotics (see below) during JHMV infection. For transfer of virus-specific T cells, mice were intraperitoneally (i.p.) infected with $2.5 \times 10^5$ PFU the DM strain of MHV (*Stiles et al., 2006*; *Bergmann et al., 2004*; *Glass and Lane, 2003*; *Dickey et al., 2016*; *Glass et al., 2004*; *Glass and Lane, 2003*; *Plaisted et al., 2014*). At day 7 post-infection (p.i.), spleens were collected and CD4 +or CD8+ T cells were enriched using magnetic based separation columns (MACS Miltenyi Biotec). The frequencies of virus-specific T cells were determined using CD4+ and CD8+ immunodominant tetramers (PE-conjugated tetramers I-A$^b$/M133–147 and D$^b$/S510–518, respectively) at 8 µg/ml (NIH Tetramer Core Facility) (*Stiles et al., 2006*), JI; *Chen et al. (2014)*, Stem Cell Reports); $2.5 \times 10^5$ virus-specific cells were transferred via retro-orbital injection into SPF Rag1$^{-/-}$ recipient animals 3 days p.i. and brains from recipient mice were collected 7 days post-transfer to determine brain viral titers (*Dickey et al., 2016*; *Stiles et al., 2006*). Rag1$^{-/-}$ mice were randomly chosen to receive either SPF or GF T cells. Viral titers within the brains were determined on the DBT astrocytoma line using previously defined protocols.

## CNS cell isolation and flow cytometry

After animal sacrifice and perfusion with ice cold PBS, half of the brain of mice was collected and homogenized at defined times p.i. and leukocytes enriched on a Percoll gradient as previously described (*Blanc et al., 2014*). Cells were stained using the following antibodies: CD4 (GK1.5, Biolegend), CD3 (145–2 C11, Biolegend), IFN-γ (XMG1.2, Biolegend), FoxP3 (FJK-16s, eBioscience), IL-10 (JES5-16E3, Biolegend), CD8 (53-6.7, Biolegend), CD45 (30-F11, Biolegend), F4/80 (BM8, Biolegend), CD11b (M1/70, Biolegend), CD40 (HM40-3, Biolegend), CD86 (GL-1, Biolegend), MHCI (M1/42, Biolegend), and MHCII (M5/114.15.2, Biolegend). Virus-specific T cells were determined via flow cytometric analysis through either intracellular staining for IFN-γ or defined tetramers specific for immunodominant CD4 +and CD8+T cell-specific viral epitopes (*Chen et al., 2014*; *Stiles et al., 2006*). Samples were analyzed using BD LSRFortessa and FACSDiva software and data was measured using FlowJo.

## qPCR analysis

RNA was collected from cells using a Ribozol (Amresco) extraction method. RNA was treated with DNaseI (Sigma-Aldrich) per supplier protocol. cDNA was synthesized using qScript cDNA SuperMix (Quantigene). Amplification of cDNA was measured using Lightcycler 480 SYBR Green I Master (Roche) with the CT method and normalized to L32. Primers used were L32F 5-AAGCGAAAC TGGCGGAAAC-3, L32R 5-TAACCGATGTTGGGCATCAG-3, CIITAF 5-ACACCTGGACCTGGAC TCAC-3, CIITAR 5-GCTCTTGGCTCCTTTGTCAC-3, TGF-BF 5-CTGCTGAGGCTCAAGTTAAAAGTG-3, TGF-BR 5-CAGCCGGTTGCTGAGGTA-3. JHMV-F 5-CGAGCCGTAGCATGTTTATCTA-3, JHMVR 5-CGCATACACGCAATTGAACATA-3. TLR4F 5-TGGGTCAAGGAACAGAAGCAGT-3 TLR4R 5-AA TCCAACACTAAGGAGGTATTCA-3.

## CD11b + cell enrichment

Enrichment of CD11b + cells was performed by subjecting single-cell suspension of brain homogenates (as performed in CNS Cell Isolation and Flow Cytometry) to CD11b+ (microglia) Microbead MACS purification (Miltenyi Biotec) according to manufacture instruction.

## Microglia isolation and analysis

After animal sacrifice at defined times p.i. and leukocytes enriched on a Percoll gradient as previously described (*Blanc et al., 2014*). Cell were sorted at the University of Utah Flow Cytometry Core based using the following markers, CD45$^{lo}$, CD11b+, F4/80+. Microglia were then plated in a 96-well flat bottom plate coated with Poly-D-Lysine (Sigma Aldrich) and incubated with the S510 peptide overnight. Spleens were collected from MHV-DM infected SPF mice 7 D.P.I., viral-specific CD8 +T cells were sorted at the University of Utah flow cytometry core. T cells were stained using Far Red Cell Trace or CFSE (Thermo Fisher), CD8 +T cells were then co-cultured with microglia for 72 hr, samples were analyzed using BD LSRFortessa and FACSDiva software and data was measured using FlowJo.

## Histology

Experimental mice were sacrificed at defined times points and the length of spinal cord extending from thoracic vertebrate 6–10 was cryoprotected in 30% sucrose, cut into 1 mm transverse blocks and processed so as to preserve the craniocaudal sequence/orientation and subsequently embedded in O.C.T. (VWR, Radnor, PA). Eight-µ M-thick coronal sections were cut and sections were stained with hematoxylin/eosin (H &E) in combination with luxol fast blue (LFB) and between 4 and 8 sections/mouse analyzed. Areas of total white matter and demyelinated white matter were determined with Image J Software and demyelination was scored as a percentage of total demyelination from spinal cord sections analyzed as previously described (*Blanc et al., 2015*; *Blanc et al., 2014*; *Dickey et al., 2016*).

## Antibiotic treatment and TLR ligands

Breeder pairs of Adult SPF mice were treated with an antibiotic cocktail of Ampicillin, Erythromycin, Neomycin, and Gentamycin (0.5 g/L) supplemented with Equal (4 g/L), progeny of these breeder pairs were also kept on the same antibiotic cocktail after weaning. For TLR ligand experiments, Pam3CSK4 (10 µg/mL) (Invivogen) and LPS (10 µg/mL) (Sigma-Aldrich) were added to the drinking water of germfree mice for 2 weeks prior to infection. Germfree mice were chosen at random to receive TLR ligands or to serve as controls.

## BV-2 cells

The murine BV-2 cell line was acquired from the ATCC and was mycoplasma-free grown in RPMI and supplemented with SCFAs [Acetate, Butyrate, Propionate (0.1 µg/mL), Sigma Aldrich] TLR ligands [LPS, Pam3CysK4, FSL-1 and Poly I:C (0.1 µg/mL), Invivogen] and recombinant mouse cytokines [IL-1 IL-18, and IL-33 (0.25 µg/mL), Biolegend] for 24 hr.

## Tamoxifen treatment

At age of weaning (3 weeks), mice were orally gavaged with 0.2 mg tamoxifen per g body weight once daily for 5 days as previously described (Fonseca et al Journal of Neuroinflammation 201714:48). Mice were given 1 month after tamoxifen administration to allow for myeloid cell turnover before infection with JHMV.

## TLR2 and 4 detection

HEK-Blue mTLR2 or mTLR4 cells (InvivoGen) were used according to manufacturer's instruction to determine serum concentrations of TLR2 and TLR4 and if JHMV signals through TLR4. After passaging according to manufacturer instruction, a flat-bottom 96-well plate was loaded with 25,000 cells per well. Then, 20 µL of PBS, mouse serum, varying dilutions of JHMV or serial dilutions of 0.1 µg/ml LPS added to each well. Cells were incubated for 6 hr, spun down and supernatant isolated. 20 µL of supernatant was then incubated with 180 µL of QUANTI-Blue solution (InvivoGen) for 15 min. Optical density at 620 nm was then measured using a Biotek Synergy H1M plate reader.

## Sample size estimation

No a priori power analyses were used to determine sample size. Sample size was determined by availability of germ-free mice and age/sex matched controls (when applicable). The sample sizes reported here are similar to those in previous literature utilizing the JHMV model (*Ireland et al., 2008*). Sample sizes are provided within the legend of each figure.

## Statistics

Pair-wise comparison of experimental groups were performed with an unpaired two-tailed Student' t-test or one-way ANOVA or Mann-Whitney test for viral titers. JHMV disease curves were analyzed using two-way ANOVA or area under the curve analysis. Outlier testing was performed using Grubb's test for outliers. If an outlier was detected (p<0.05), that data point was omitted from any analysis. All statistics were performed using Prism six software (GraphPad Software).

# Additional information

## Funding

| Funder | Grant reference number | Author |
|---|---|---|
| Multiple Sclerosis Society | Center Grant | Thomas E Lane |
| Burroughs Wellcome Fund | Investigator of Pathogenesis | June L Round |
| National Institutes of Health | T32HD007491 | D Garrett Brown |
| National Institutes of Health | R01AG047956 | Ryan M O'Connell |
| Ben B. and Iris M. Margolis Foundation | | June L Round<br>Ryan M O'Connell<br>Thomas E Lane |
| National Institutes of Health | GM007464 | Raymond Soto |
| National Institutes of Health | AI055434 | Charisse Petersen |
| National Institutes of Health | R01AI123106-04 | Ryan M O'Connell |
| National Institutes of Health | DP2AT008746-01 | June L Round |

The funders had no role in study design, data collection and interpretation, or the decision to submit the work for publication.

## Author contributions

D Garrett Brown, Raymond Soto, Conceptualization, Formal analysis, Investigation, Methodology; Soumya Yandamuri, Investigation, Methodology; Colleen Stone, Laura Dickey, Formal analysis, Investigation, Methodology; Joao Carlos Gomes-Neto, Elissa D Pastuzyn, Rickesha Bell, Charisse Petersen, Kaitlin Buhrke, W Zac Stephens, Investigation; Robert S Fujinami, Ryan M O'Connell, Jason D Shepherd, Conceptualization; Thomas E Lane, Conceptualization, Funding acquisition, Project administration; June L Round, Conceptualization, Formal analysis, Funding acquisition, Methodology, Project administration

## Author ORCIDs

D Garrett Brown (ID) https://orcid.org/0000-0001-8779-7048
Raymond Soto (ID) https://orcid.org/0000-0002-8483-6459
Rickesha Bell (ID) http://orcid.org/0000-0002-0554-0730
Jason D Shepherd (ID) http://orcid.org/0000-0001-7384-8289
Thomas E Lane (ID) https://orcid.org/0000-0003-0392-0825
June L Round (ID) https://orcid.org/0000-0002-7158-9874

## Ethics

Animal experimentation: This study was performed in strict accordance with the recommendations in the Guide for the Care and Use of Laboratory Animals of the National Institutes of Health. All of the animals were handled according to approved institutional animal care and use committee (IACUC) protocol # 17-04009 at the University of Utah.

## Decision letter and Author response

Decision letter https://doi.org/10.7554/eLife.47117.021
Author response https://doi.org/10.7554/eLife.47117.022

# Additional files

## Supplementary files

• Supplementary file 1. Key resources table.
DOI: https://doi.org/10.7554/eLife.47117.018
• Transparent reporting form

DOI: https://doi.org/10.7554/eLife.47117.019

**Data availability**

All data generated during this study are included in the manuscript and supporting files.

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
