## [Decision Letter]

Thank you for submitting your article "The gut microbiota protects from viral-induced neurologic damage through microglia-intrinsic TLR signaling" for consideration by *eLife*. Your article has been reviewed by two peer reviewers, and the evaluation has been overseen by a Reviewing Editor and Wendy Garrett as the Senior Editor. The following individuals involved in review of your submission have agreed to reveal their identity: Dan Carr (Reviewer #1); Susan R Weiss (Reviewer #2).

The reviewers have discussed the reviews with one another and the Reviewing Editor has drafted this decision to help you prepare a revised submission.

Summary:

Brown et al., have used a murine coronavirus model of central nervous system (CNS) disease, primarily demyelination, to investigate the role of gut microbiota to protect from neurologic damage. This is a thorough study in which much of the data are derived from comparing wild type specific pathogen free (SPF) mice with germ-free (GF) mice as well as SPF mice treated with antibiotics providing two models of comparing disease in mice with a 'normal' microbiota with those microbiota free or with reduced microbiota. The data in this manuscript support the authors’ conclusion that signals from microbiota can activate microglia via TLR4, to increase microglial antigen presentation and subsequent antiviral T cells and protect from virus-induced CNS pathogenesis.

While the impact of the microbiota on the immune system and the immune response to infection has been described for a variety of pathogen insults, the current undertaking addresses an understudied area – CNS infection and neuroimmunology. Together, the results described herein provide a clear demonstration of the importance of the gut microbiota and a strong foundation for understanding the mechanistic basis through which microbial factors impact CNS disease.

Essential revisions:

The reviewers did not feel that additional experiments are mandatory prior to resubmission. However, the did raise a number of points requiring clarification and discussion prior to publication.

1) Figure 1 is quite telling as to the impact of the microbiota of the host in minimizing damage associated with MHV infection of the CNS. The authors seem to associate the pathology with the significant reduction in Treg cells residing in the brain. In the latter part of the paper, there is reasonable effort to clearly associate the performance of effector Th1 cells and microglia but there is no explanation as to why there is an early increase in Tregs residing in the brain (day 7 p.i.) and a loss in this cell population during the chronic stages of disease. Also, do germ-free mice ever clear the viral pathogen from the brain as shown in panel B?

2) In reference to Figure 1—figure supplement 1, although it is likely antibiotic treatment of SPF breeder mice significantly reduced the microbiota in the gut, there is no evidence shown as to how much of the bacterial burden was reduced.

3) Figure 2 is an important addition to the Results section and demonstrates the critical component to be independent of effector T cell performance in the sensitivity of germ-free mice to MHV CNS infection.

4) Defining the aberrant immune response in germ-free animals to MHV infection to be at the level of the microglia in results presented in Figures 3-4 especially with the use of the PLX-3397 reagent. A subpopulation of dendritic cells is also found to express CSF1R. Was there a loss of CSF1R-expressing cells in organized lymphoid tissue (e.g., spleen or cervical lymph nodes) in PLX-3397-treated mice? Does the loss of PLX-3397 result in an alteration in the number and frequency of Treg cells in the brain thus, providing more information as to demyelination, loss of microglia, and reflected changes in the Treg cell population as observed in Figure 1?

5) In Figure 5C and Figure 7A, is there a reason why the authors stopped the experiment at day 9 p.i.? It seems to be inconsistent with that reported in Figure 1 in which the clinical disease does not reach a plateau until 12 days p.i.

6) The role of TLR4 ligands in reducing the exacerbation of neurologic disease in germ-free mice is impressive. One aspect the authors considered was whether there was a direct activation of TLR4 by JHMV using a reporter line. However, it is unclear whether JHMV killed the HEK-blue reporter cells and therefore, lead to spurious results as reported in Figure 6—figure supplement 1.

7) The results included in Figure 7 strongly underscore the critical role of TLR4 in microglia in reducing the severity of disease following JHMV CNS infection. As a side note, it is also nice to see confirmation of nestin expressing cells do not differentiate to microglia.

8) The murine coronavirus JHM model is well characterized and appropriate for the questions asked. However, since there are many variants of JHM, the authors should briefly describe the pathogenesis of the strain used (JHMV-34; while the publication referenced uses JHMV strain J2.2v-1) and the scale they are using for clinical signs. More specifically does this strain cause damage other than demyelination and is the clinical score a reflection only or mainly of demyelination? Furthermore, microglia are both protective and also a target for viral infection; how does deleting this target of infection effect the pathogenesis? More explanation is needed also of the "DM-MHV" strain that is used for immunization.

9) Using a mouse microglia cell line to screen the authors identified TLR ligands as the source of priming microglia and found that feeding these ligands to mice resulted in increased T cells in the CNS and reduced disease. The authors should briefly describe the BV2 microglia cell line. They further showed using KO mice and that TLR4 signaling by microglia as a requirement for virus induced demyelination since KO of TLR4 (either global or in in microglia) but not in nestin expressing CNS cells only lead to increased CNS disease.

10) Re: Figure 5, it is not stated whether there was any difference in response between mice fed LPS and Pam3CysK and LPS alone? Also, it is stated in the subsection “Oral administration of TLR ligands is sufficient to enhance microglia activation and protect from neurologic disease”, that high levels, but not low levels of cytokines etc. could promote increased frequency of MHCII on BV2 cells. The levels actually used in vitro and in vivo are somewhat unclear. Materials and methods describes concentrations ug/ml while Figure 5—figure supplement 1A and B (titration) is in molarity. What concentration is used in Figure 5B and Figure 5—figure supplement 1C?

---

## [Author Response]

Essential revisions:The reviewers did not feel that additional experiments are mandatory prior to resubmission. However, the did raise a number of points requiring clarification and discussion prior to publication.1) Figure 1 is quite telling as to the impact of the microbiota of the host in minimizing damage associated with MHV infection of the CNS. The authors seem to associate the pathology with the significant reduction in Treg cells residing in the brain. In the latter part of the paper, there is reasonable effort to clearly associate the performance of effector Th1 cells and microglia but there is no explanation as to why there is an early increase in Tregs residing in the brain (day 7 p.i.) and a loss in this cell population during the chronic stages of disease. Also, do germ-free mice ever clear the viral pathogen from the brain as shown in panel B?

Differential infiltration of regulatory T cells (Tregs) into the CNS of JHMV infected GF during early (day 7) and later (day 21) stages of disease most likely reflects differences in expression of proinflammatory cytokines/chemokines expressed at these stages of disease that serve to attract Tregs into the CNS. Increases in Tregs of GF mice were also reported during the EAE model (Lee et al., 2011). We have not looked past day 21 p.i. to determine if infected GF mice ultimately reduce viral titers below the level of detection. Focused future studies will be necessary to carefully understand a role for microbiota induced Tregs in this model. We have changed our wording to soften the claims that our phenotype is due to changes in Tregs.

2) In reference to Figure 1—figure supplement 1, although it is likely antibiotic treatment of SPF breeder mice significantly reduced the microbiota in the gut, there is no evidence shown as to how much of the bacterial burden was reduced.

The same antibiotic course was utilized in Soto et al., 2017 (Soto et al., 2017) in which bacterial populations were reduced approximately 100-fold. We have added this reference to the manuscript.

3) Figure 2 is an important addition to the Results section and demonstrates the critical component to be independent of effector T cell performance in the sensitivity of germ-free mice to MHV CNS infection.

We thank the reviewer for the positive comments on the data presented in Figure 2.

4) Defining the aberrant immune response in germ-free animals to MHV infection to be at the level of the microglia in results presented in Figures 3-4 especially with the use of the PLX-3397 reagent. A subpopulation of dendritic cells is also found to express CSF1R. Was there a loss of CSF1R-expressing cells in organized lymphoid tissue (e.g., spleen or cervical lymph nodes) in PLX-3397-treated mice? Does the loss of PLX-3397 result in an alteration in the number and frequency of Treg cells in the brain thus, providing more information as to demyelination, loss of microglia, and reflected changes in the Treg cell population as observed in Figure 1?

The reviewer brings up an important and relevant point regarding CSFR1 being expressed on dendritic cells (DCs) as Klein and colleagues (Funk and Klein, 2019) clearly showed diminished numbers of these cells in secondary lymphatic tissues following PLX5622-treatment of mice infected with West Nile Virus. In this study, we provide data that CD45+Cdllb+ cells are not diminished in the spleen of PLX3392 treated animals (Figure 4—figure supplement 1), however we did not look at the effects of PLX3397 treatment on sub-populations of DCs within either the brain or secondary lymphatic tissue. Regarding Tregs, we did not evaluate the how PLX3397 influenced Treg infiltration into the CNS of MHV-infected mice. We intend to carry out a focused study on microbiota induced Tregs in future studies using this model.

5) In Figure 5C and Figure 7A, is there a reason why the authors stopped the experiment at day 9 p.i.? It seems to be inconsistent with that reported in Figure 1 in which the clinical disease does not reach a plateau until 12 days p.i.

In both of these experimental set-ups some of the mice in one group were reaching our criteria for endpoint (weight loss greater than 15% and/or moribund). Therefore, in these experiments we had to take all the mice at this time point to acquire a thorough analysis.

6) The role of TLR4 ligands in reducing the exacerbation of neurologic disease in germ-free mice is impressive. One aspect the authors considered was whether there was a direct activation of TLR4 by JHMV using a reporter line. However, it is unclear whether JHMV killed the HEK-blue reporter cells and therefore, lead to spurious results as reported in Figure 6—figure supplement 1.

We did not assess cell viability after coculture with JHMV. However it is unlikely that JHMV killed the HEK-blue reporter cells as MHV strains do not infect human cells. In further support of this, the human protein atlas database also finds little to no expression of CEACAM1 (the receptor for JHMV) on HEK293 cells also

(https://www.proteinatlas.org/ENSG00000079385-CEACAM1/cell). Regardless, this alternative hypothesis cannot be currently discounted with complete certainty.

7) The results included in Figure 7 strongly underscore the critical role of TLR4 in microglia in reducing the severity of disease following JHMV CNS infection. As a side note, it is also nice to see confirmation of nestin expressing cells do not differentiate to microglia.

We thank the reviewer for the positive comments on the data presented in Figure 7.

8) The murine coronavirus JHM model is well characterized and appropriate for the questions asked. However, since there are many variants of JHM, the authors should briefly describe the pathogenesis of the strain used (JHMV-34; while the publication referenced uses JHMV strain J2.2v-1) and the scale they are using for clinical signs. More specifically does this strain cause damage other than demyelination and is the clinical score a reflection only or mainly of demyelination? Furthermore, microglia are both protective and also a target for viral infection; how does deleting this target of infection effect the pathogenesis? More explanation is needed also of the "DM-MHV" strain that is used for immunization.

We appreciate the reviewer’s comments on use of MHV to evaluate the effects of the gut microbiome on viral-induced disease. In addition, we acknowledge the reviewer’s concerns on the numerous MHV strains employed by investigators within the field and nomenclature differences. In brief, for this study we used the JHMV strain J2.2v-1 which we have used in the majority of studies published from the Lane laboratory over the previous 20 years. Following intracranial inoculation with this virus into C57BL/6 mice, animals will develop an acute encephalomyelitis characterized by widespread infection of glial cells (with relative sparing of neurons) followed by an immune-mediated demyelinating disease characterized by viral persistence in white matter tracts, glial activation accompanied by infiltration of activated T cells and macrophages into the CNS. The DM-MHV strain of virus was originally developed in the laboratories of Dr. Stephen Stohlman and Dr. Cornelia Bergmann and provided to Dr. Lane. Dr. Lane, along with Drs. Stohlman and Bergmann (Bergmann et al., 2004) have used this virus for induction of robust T cell responses following peripheral injection (Dickey et al., 2016; Glass et al., 2004; Glass and Lane, 2003; Plaisted et al., 2014). We have clarified these issues in both the Results section and Materials and methods and provided additional references.

9) Using a mouse microglia cell line to screen the authors identified TLR ligands as the source of priming microglia and found that feeding these ligands to mice resulted in increased T cells in the CNS and reduced disease. The authors should briefly describe the BV2 microglia cell line. They further showed using KO mice and that TLR4 signaling by microglia as a requirement for virus induced demyelination since KO of TLR4 (either global or in in microglia) but not in nestin expressing CNS cells only lead to increased CNS disease.

The BV2 cell line was originally developed in 1990 by infecting primary microglial cell cultures with a -*raf*/v-*myc* oncogene carrying retrovirus (Blasi et al., 1990) and has been used by numerous investigators as an effective alternative model system for primary microglia cultures study various cell biology aspects (Henn et al., 2009). We have now included this information in the revised Results section.

10) Re: Figure 5, it is not stated whether there was any difference in response between mice fed LPS and Pam3CysK and LPS alone? Also, it is stated in the subsection “Oral administration of TLR ligands is sufficient to enhance microglia activation and protect from neurologic disease”, that high levels, but not low levels of cytokines etc. could promote increased frequency of MHCII on BV2 cells. The levels actually used in vitro and in vivo are somewhat unclear. Materials and methods describes concentrations ug/ml while Figure 5—figure supplement 1A and B (titration) is in molarity. What concentration is used in Figure 5B and Figure 5—figure supplement 1C?

Differences were observed when feeding GF mice LPS alone or in combination with Pam3CysK. In both cases, we observed more leukocytes in the brains of infected mice. We also saw more CD4+ T cells in both cases. However, only when feeding both ligands did we observe increased frequency of CD4+ and CD8+ T cells and increased numbers of CD8+ T cells. When feeding mice only LPS, we observed increased number and frequency of MHCII+ microglia, this was not observed when feeding both ligands, however. The text has been altered to address this.

Figure 5B and Figure 5—figure supplement 1 report the effects of treating microglia with 0.10 μg/mL of each SCFA or TLR ligand or 0.25 μg/mL of cytokine. Legends for these Figures have been modified to address this.